# Influence of Drought Stress on the Rhizosphere Bacterial Community Structure of Cassava (*Manihot esculenta* Crantz)

**DOI:** 10.3390/ijms25137326

**Published:** 2024-07-03

**Authors:** Huling Huang, Mingchao Li, Qiying Guo, Rui Zhang, Yindong Zhang, Kai Luo, Yinhua Chen

**Affiliations:** 1School of Breeding and Multiplication, Sanya Institute of Breeding and Multiplication, Hainan University, Sanya 572025, China; 22220951310162@hainanu.edu.cn (H.H.); 22210901000021@hainanu.edu.cn (M.L.); 21220951310020@hainanu.edu.cn (Q.G.); zhangrui@hainanu.edu.cn (R.Z.); 2School of Tropical Agriculture and Forestry, Hainan University, Haikou 570228, China; 3Key Laboratory of Plant Disease and Pest Control of Hainan Province, Institute of Plant Protection, Hainan Academy of Agricultural Sciences, Haikou 571100, China; 23300558@163.com

**Keywords:** drought stress, cassava, rhizosphere bacterial community, environmental factor, drought response

## Abstract

Drought presents a significant abiotic stress that threatens crop productivity worldwide. Rhizosphere bacteria play pivotal roles in modulating plant growth and resilience to environmental stresses. Despite this, the extent to which rhizosphere bacteria are instrumental in plant responses to drought, and whether distinct cassava (*Manihot esculenta* Crantz) varieties harbor specific rhizosphere bacterial assemblages, remains unclear. In this study, we measured the growth and physiological characteristics, as well as the physical and chemical properties of the rhizosphere soil of drought-tolerant (SC124) and drought-sensitive (SC8) cassava varieties under conditions of both well-watered and drought stress. Employing 16S rDNA high-throughput sequencing, we analyzed the composition and dynamics of the rhizosphere bacterial community. Under drought stress, biomass, plant height, stem diameter, quantum efficiency of photosystem II (Fv/Fm), and soluble sugar of cassava decreased for both SC8 and SC124. The two varieties’ rhizosphere bacterial communities’ overall taxonomic structure was highly similar, but there were slight differences in relative abundance. SC124 mainly relied on Gamma-proteobacteria and Acidobacteriae in response to drought stress, and the abundance of this class was positively correlated with soil acid phosphatase. SC8 mainly relied on Actinobacteria in response to drought stress, and the abundance of this class was positively correlated with soil urease and soil saccharase. Overall, this study confirmed the key role of drought-induced rhizosphere bacteria in improving the adaptation of cassava to drought stress and clarified that this process is significantly related to variety.

## 1. Introduction

Drought represents a significant challenge to global food security by adversely impacting crop development and yield by negatively affecting the quality of morphological, physiological, and biochemical aspects of plants [1]. The frequency, intensity, and duration of drought events are anticipated to rise due to global climate change and dramatically reduce crop yields, further exacerbating the negative impact on agricultural production worldwide [2,3]. Given the growing demand for food production and security, the development of effective eco-friendly strategies to enhance plant resilience to water deficits and improve crop growth and yield is imperative [4]. Plant health is linked to the activity of associated microbes, with plants influencing the composition of their rhizosphere microorganism [5]. Manipulating plant-associated microbiomes presents an effective and environmentally sustainable strategy to enhance plant resistance and mitigate the adverse effects of drought stress on agricultural productivity [6,7]. The rhizosphere, a microecological zone connecting plant roots and soil, is pivotal for plant–soil microbe interactions [8,9,10], influencing host plant development [11,12,13], nutrient uptake [14,15], pathogen immunity, and abiotic stress tolerance [6,14,16,17,18,19]. Mycorrhizae are beneficial microbes that play a crucial role in supporting plant nutrition, promoting plant development, and enhancing plant health [20]. Plants and arbuscular mycorrhizal fungi (AMF) exist in a symbiotic relationship. The uptake of crucial nutrients such as potassium, nitrogen, calcium, phosphorus, and magnesium improve when crop roots are inoculated with mycorrhiza. This symbiosis between plants and AMF can enhance plant growth and improve plant resistance to abiotic stress [21]. Zhang et al. [22] established AM symbiosis between tomato (*Solanum lycopersicum*) plants and three AMF species (*Rhizophagus intraradices*, *Funneliformis mosseae*, *Rhizophagus irregularis*) under well-watered and drought-stressed conditions in a pot experiment. The results demonstrated that AMF significantly enhanced plant growth, with varying growth-promoting ability and drought tolerance of the three fungi. Additionally, they also found that AMF mediated plant drought tolerance by regulating both AM-specific and non-AM-specific fatty acids (FAs) in a complementary manner. Beneficial rhizobacteria enhance mineral nutrient uptake and assimilation, modulate phytohormone levels, induce antioxidant defense systems, and contribute to osmotic adjustment, thereby supporting plant adaptability to environmental stresses [23]. Therefore, understanding the role of rhizobacteria under drought stress and their assembly according to their functions can maximize the power of microbial communities to support plant resistance to drought [24].

Plants adopt multiple strategies to navigate the challenges of drought stress, which include a combination of stress avoidance and regulation of drought tolerance, depending on their genotypic characteristics. Specifically, drought-resistant plants have developed a repertoire of strategies for drought adaptation that differ significantly from those of drought-sensitive plants [25], enabling them to swiftly initiate drought tolerance responses at both physiological and molecular levels upon exposure to drought stress. Moreover, drought stress influences host plants and surrounding soil, thereby altering the structure and function of rhizosphere microbial communities, with implications for plant performance and health [26,27,28,29,30]. The composition of the rhizosphere microbial community under drought conditions is influenced by both the host plant’s properties and the characteristics of the surrounding soil environment [31]. Considering the variability in drought tolerance among different communities of rhizosphere microbes, the “cry for help” hypothesis posits that certain distinct genotypes of host plants can selectively utilize specific bacterial through the release of particular root exudates [32]. Plants recruit specific microbes through a so-called “cry for help” mechanism to enhance stress tolerance [33]. Thus, plants with certain genotypes significantly influence rhizosphere microbiome diversity, which is crucial for host adaptation to environmental stresses, including drought. 

Cassava (*Manihot esculenta* Crantz) is a crucial tropical crop for food and industrial material [34]. Despite its drought tolerance, cassava’s productivity is significantly reduced under insufficient rainfall conditions, presenting a challenge to starch production enhancement [35]. Recent studies have primarily focused on identifying drought-responsive genes and proteins in cassava [36,37,38,39]. Investigations of morphological, physiological, and agronomic traits can help us understand the molecular mechanisms of drought stress tolerance in cassava [37,40,41]. A previous study, by analyzing the composition and diversity of bacterial communities associated with the rhizospheres of fourteen cassava varieties, revealed that the rhizosphere microbial communities were relevant to the plants with certain genotypes [42]. The authors hypothesized that the diversity of the rhizosphere microbial community of different cassava varieties resulted from the long-term selection and shaping of the rhizosphere environment by the host cassava, which may help it adapt to adversity. Na et al. [43] investigated the changes in rhizosphere microbial community structure of two millet (*Panicum miliaceum*) varieties under drought stress. Their study found that while drought stress did not significantly affect the overall diversity and structure of the bacterial rhizosphere community between the two cultivars, there were differential responses in specific bacterial taxa, such as *Actinobacteria*, *Acinetobacter*, *Lysobacter*, *Streptomyces*, and *Cellvibrio*. Similarly, Li et al. [44] used 16S rRNA gene amplicon sequencing and widely targeted metabolomic analysis to examine the rhizosphere soil and root exudates from two contrasting rice (*Oryza sativa*) varieties, Nipponbare and Luodao 998, under drought stress. Their findings highlighted plants with certain genotype-specific responses to drought at the microbial and metabolic levels. Additionally, Zoppellari et al. [45] demonstrated that inoculating maize (*Zea mays*) with beneficial microbes could enhance the plant’s drought resistance, emphasizing the potential of microbial interventions in improving crop resilience to drought stress. However, the role of rhizosphere microbes in stress resistance remains less explored in cassava, compared to other plants [46]. 

In this present study, we examined two cassava varieties (SC8—sensitive, and SC124—relative tolerant) with drought responses through 16S rDNA deep sequencing analysis of their rhizosphere bacterial communities under both drought and well-watered conditions. Our objectives were to (I) assess the impact of drought and variety on the diversity and composition of rhizosphere bacterial communities; (II) identify bacterial taxa potentially linked to increased drought stress tolerance; and (III) explore the relationship between bacterial community structure and soil environmental factors. This research aims to shed light on the assembly process of host plants to rhizosphere microorganisms under drought stress in host plants and facilitate the development of rhizosphere bacteria application protocols to improve drought stress resistance in cassava.

## 2. Results

### 2.1. Effects of Drought and Variety on Soil Chemical Properties and Enzyme Activities

Drought stress can impact soil nutrient content and enzyme activity. This study found that drought stress significantly decreased soil volume water content, total nitrogen, total phosphorus, available phosphorous, and soil urease (S-UE), while pH, solid acid phosphatase (S-ACP), and soil saccharase (S-SC) showed no significant difference. Under drought-stress conditions, the total organic carbon content of the sensitive variety SC8 was significantly decreased, whereas that of the drought-tolerant variety SC124 remained stable. Conversely, solid catalase (S-CAT) activity significantly decreased in SC124. The total nitrogen and total organic carbon contents of SC124 were higher than those of SC8 under drought stress (Table 1).

### 2.2. Effects of Drought and Variety on Plant Phenotypes and Physiological State

The plant properties of the two cassava varieties, SC8 and SC124, showed differences under both well-watered and drought conditions. After exposure to drought stress, reductions were in shoot biomass, root biomass, plant height, stem diameter, quantum efficiency of photosystem II (Fv/Fm), and relative chlorophyll content (SPAD) in both varieties (Figure 1 and Figure 2). Notably, SC8 showed a significant reduction in shoot biomass, root biomass, plant height, and Fv/Fm between the well-watered and drought conditions, whereas for SC124, significant changes were observed in shoot biomass, plant height, and Fv/Fm. In particular, SC124 exhibited higher plant height, stem diameter, Fv/Fm, and SPAD under both conditions compared to SC8. MDA content, an indicator of oxidative stress, increased significantly in SC124, contrasting with a significant decrease in SC8. Soluble sugar, crucial for osmoregulation, decreased after drought stress, with a notable difference in SC8 compared to the well-watered condition, whereas the change was not significant in SC124. In addition, the proline content in SC124 was significantly lower than that in SC8 across both conditions (Figure 2). Overall, SC8 faced more pronounced reductions in morphological and physiological indices under drought, which confirms the worse drought tolerance. 

### 2.3. Effects of Drought and Variety on Rhizosphere Bacterial Communities

The rhizosphere soil from samples of drought-resistant and sensitive cassava varieties was sequenced using 16S rDNA amplicon sequencing technology. A total of 960,535 original bacteria sequences were detected. The highest number of effective sequences was found in SC8-D (mean 71,589), while the lowest was in SC8-C (mean 70,536). The proportion of high-quality sequences in each sample exceeded 88%, and the coverage index was above 0.98 (Appendix A). 

The alpha diversity of the cassava rhizosphere bacterial communities was assessed using ACE, Chao1, Simpson, and Shannon indices, revealing significant differences between SC124 and SC8 under both well-watered and drought conditions, with SC8’s rhizosphere showing greater bacterial diversity (Table 2). There were also significant differences in bacterial richness (ACE, Chao1) between the well-watered and drought conditions for both varieties, alongside significant variations in the diversity indices among the treatments (Table 2), indicating diverse bacterial presence.

Operational Taxonomic Units (OTUs) were classified based on a similarity level above 97% (Appendix A). The rarefaction curve of each sample tended to be flat with the increase of sequencing data, confirming the adequacy and accuracy of the sequencing effort. Venn diagrams highlighted the distribution of OTUs across treatments, totaling 7776 OTUs. The number of OTUs specific to SC124-C, SC124-D, SC8-C, and SC8-D were 607, 420, 583, and 326, respectively. The number of unique OTUs in drought-resistant variety SC124 was higher under both conditions, suggesting that the drought-resistant variety harbored more specific bacterial species (Figure 3). Principal coordinates analysis (PCoA) based on the weighted and unweighted UniFrac distance showed that samples from each treatment clustered together, yet the overall distribution was dispersed (Figure 4). Both SC124 and SC8 varieties were significantly influenced by drought stress, as illustrated by the distinct distribution of communities on the principal coordinate axes. 

Taxonomic classification of rhizosphere soil bacteria in different cassava varieties under drought stress was shown in Appendix A. A phylum-based population structure analysis of the rhizosphere bacteria under different treatments revealed the dominant microflora in the rhizosphere based on their relative abundance (Figure 5A). The top three relatively abundant phyla were Proteobacteria, Actinobacteria, and Acidobacteriota. In well-watered conditions, SC124’s rhizosphere significantly enriched Proteobacteria (36.60%), Actinobacteriota (18.56%), and Acidobacteriota (9.5%), whereas under drought conditions, the abundance of Actinobacteriota decreased to 15.52%, with Acidobacteria increasing to 11.85% and Proteobacteria to 40.23%. Similarly, the well-watered rhizosphere of SC8 significantly enriched Proteobacteria (39.80%) and Actinobacteriota (14.00%), but under drought conditions, Proteobacteria slightly decreased to 39.66% and Actinobacteriota increased to 16.81% (Figure 5A). Compared with the initial soil, the relative abundance of Proteobacteria and Actinobacteriota increased under well-watered and drought stress, while the relative abundance of Bacteroidota decreased in SC8 and SC124.

Class-based population structure analysis of the rhizosphere bacteria under different conditions revealed the dominant microflora in the rhizosphere environment based on their relative abundance (Figure 5B and Figure 6). The main groups were Alpha-proteobacteria, Gamma-proteobacteria, Actinobacteria, and Bacteroidia. Under well-watered conditions, the species compositions of the rhizosphere bacterial communities of SC124 and SC8 were similar, yet their abundances differed notably (Figure 5B). Compared with the initial soil, the relative abundance of Alpha-proteobacteria increased under well-watered and drought stress, while the relative abundance of Bacteroidia and Gamma-proteobacteria decreased in SC8 and SC124. Specifically, in the drought condition for SC8, the relative abundance of Gamma-proteobacteria and Actinobacteria increased, whereas Alpha-proteobacteria and Saccharimonadia decreased. In the drought condition for SC124, the relative abundances of Alpha-proteobacteria, Gamma-proteobacteria, and Bacteroidia increased, while those of Actinobacteria decreased (Figure 5B). Additionally, after a period of drought, the relative abundance of Acidimicrobiia, Gamma-proteobacteria, Polyangia, and Acidobacteriae increased in both SC124 and SC8, with the drought-tolerant variety SC124 showing higher levels than the drought-sensitive variety SC8. Furthermore, the relative abundance of Alpha-proteobacteria and Vicinamibacteria increased significantly in SC124 but decreased significantly in SC8 under drought stress (Figure 6).

Although the shift in abundance of the cassava microbiome due to drought and variety was observed, and a group of abundant taxa in rhizosphere soils were identified, biomarkers in rhizosphere soils remain unidentified. Thus, we sought to identify microbes as biomarkers that could explain the observed differences between drought treatments or varieties by performing linear discriminant analysis effect size (LEfSe), identifying microbes with an LDA score of 4 as biomarkers (Figure 7; Appendix A). The results showed that the rhizosphere microbes we identified as biomarkers for cassava differed significantly under well-watered and drought conditions. Moreover, different varieties had their specific biomarkers under different treatment conditions. Notably, p__Acidobacteriota and p__Patescibateria were considered potential biomarkers in well-watered conditions for SC124 and SC8, respectively, while under drought conditions, p__Acidobacteriota and c__Bacteroidia had high LDA scores in rhizosphere soils of SC124 and SC8, respectively (Figure 7). 

### 2.4. Correlation Analysis between Rhizosphere Bacterial Community and Physicochemical Properties

To ascertain the environmental factors and morphological physiology influencing the compositional differences in the cassava rhizosphere microbiome, correlations between the richness and Shannon index with plant morphological physiology and soil properties were analyzed (Figure 8A). The Mantel test revealed a significant correlation of total organic carbon, available phosphorous, total phosphorus, total nitrogen, soluble sugar, Fv/Fm, and plant height with the richness index, while total phosphorus and available phosphorous were significantly correlated with the Shannon index. RDA results indicated that total nitrogen, total phosphorus, and available phosphorous were negatively corrected with S-ACP. In contrast, positive correlations were found between S-CAT and total organic carbon, total nitrogen, total phosphorus, available phosphorous, S-SC, and S-UE. These findings are consistent with the Mantel test results, which also showed that pH does not affect soil enzyme activity. Redundancy discriminant analysis relating the major bacteria to environmental factors revealed that drought stress accounted for 99.20% of the changes in bacterial rhizosphere, while the difference in drought tolerance between the two varieties accounted for a mere 0.05% of the changes. Alpha-proteobacteria and Actinobacteria showed a close relationship with total nitrogen, total phosphorus, available phosphorous, total organic carbon, S-SC, and S-UE, whereas Gamma-proteobacteria and Acidobacteriae were more closely related to S-ACP (Figure 8B). Correlation analysis revealed a significant relationship between alpha diversity and soil enzyme activities (Figure 8B), with ACE and Chao1 being positively correlated with S-SC, S-UE, and S-CAT, and the Shannon index and Simpson index being positively correlated with S-ACP. In addition, the richness index was positively correlated with total nitrogen, total phosphorus, available phosphorus, and total organic carbon, while the Simpson index and Shannon index were negatively correlated with them. The angular relationship in the RDA space suggested that the richness index was positively associated with Alpha-proteobacteria and Actinobacteria but negatively with Gamma-proteobacteria, Acidobacteriae, and Bacteroides. Conversely, the Shannon index correlated positively with Gamma-proteobacteria and Acidobacteriae and correlated negatively with Alpha-proteobacteria and Actinobacteria (Figure 8B).

### 2.5. Functional Prediction of Rhizosphere Bacterial Community

Functional predictions, utilizing the FAPROTAX database, were conducted to analyze the bacteria communities present in the rhizosphere soil of two varieties. The analysis revealed that a predominance of bacteria associated with chemoheterotrophy and aerobic-chemoheterotrophy (Figure 9A). Notably, after exposure to drought stress, there was a significant increase in microorganisms associated with nitrate reduction and ureolysis related to lignin degradation in both the SC8 and SC124 varieties (Figure 9B). 

## 3. Discussion

Facing the challenges of drought stress, plants can develop complex regulatory networks and involve specialized morphological structures aimed at reducing water loss, enabling them to either adapt to or resist drought conditions [47,48]. In this study, two cassava varieties with different drought resistances were subjected to drought stress until wilting was observed, indicated by a decrease in Fv/Fm, approximately 21 days after the initiation of drought treatment, which is consistent with Xing et al. [49]. Moreover, our examination of the morphological and physiological indices under both the well-watered and drought conditions revealed distinct responses between two cassava varieties with contrasting drought resistance. The water-stressed and the drought-sensitive cassava plants exhibited a heterogeneous growth pattern and significantly lower plant height and biomass under drought stress, which could be attributed to the constraints of potted growth or the natural variability within the variety. Under well-watered conditions, the SC8 variety exhibited shorter growth with more and wider leaves, whereas the SC124 variety grew taller with longer and narrower leaves. As a result, the well-watered SC124 variety had lower shoot biomass but greater plant height compared to the well-watered SC8 variety. Photosynthetic physiological parameters, acting as comprehensive responses to drought stress, serve as reliable indicators for evaluating both the degree of drought stress and the drought resistance of plants [50]. Our findings indicated that the chlorophyll content, under both well-watered and drought conditions, was higher in the drought-tolerant SC124 than in the drought-sensitive SC8, consistent with previous studies [51,52,53]. However, contrary to what might be expected, drought stress did not significantly alter the chlorophyll content in either cassava variety. This observation aligns with reports of the relative chlorophyll content stabilization under water deficiency, noted across various species [54,55,56]. Chlorophyll fluorescence is widely acknowledged as a fundamental metric for evaluating the interplay between photosynthesis and the plant growth environment. The Fv/Fm ratio, in particular, serves as a robust indicator of the photochemical efficiency of the photosynthetic apparatus, reflecting the potential efficiency of Photosystem II (PSII) [57]. Our findings corroborate previous observations of a decline in Fv/Fm values under drought conditions, a phenomenon similarly reported across various species [58,59,60,61]. Singh et al. [62] reported varietal differences in proline accumulation under the same leaf water potential in barley (*Hordeum vulgare*) varieties, which they found to be positively associated with field drought resistance. Conversely, Matos et al. [63] demonstrated that water shortage results in diminished proline content in *Cicer arietinum*. Contrary to these findings, our research indicates that drought stress does not alter the proline content, suggesting that discrepancies in crop species, treatment methodologies, and duration of stress exposure might influence outcomes. The above studies indicate that under specific drought conditions, plants with certain genotypes deploy protective mechanisms to sustain their growth requirements to the greatest extent possible.

The rhizosphere, characterized by intense interactions between crop roots, soil, and microorganisms is significantly impacted by drought, with changes in the microbial community composition being regulated by certain genotypes of host plants [7,64]. Recent research suggested that the observed disparities in drought tolerance across different plant varieties are, to a notable extent, attributable to variations within their rhizosphere microbial communities [65]. In this present study, the analysis of *α* and *β* diversity of the bacterial community highlighted differences in the bacterial composition of the cassava rhizosphere between the two varieties under both well-watered and drought conditions. Drought stress notably reduced bacterial diversity (richness index) in the rhizosphere of cassava, particularly in the drought-sensitive variety SC8, suggesting a significant correlation between changes in bacterial diversity and plant variety. These findings suggested that cassava rhizosphere microorganisms possess a degree of resistance to drought interference, but this ant-interference ability and the degree of stress or the length of time remain to be further explored. The impact of variety type on rhizosphere microorganisms has been extensively studied [66,67]. Drought-tolerant plants have a higher relative abundance of rhizosphere soil microorganisms compared to drought-sensitive plants [68], suggesting that drought-tolerant plants can recruit beneficial microorganisms to better adapt to adverse conditions [69]. However, the rhizosphere constitutes a complex environment highly sensitive to changes induced by drought stress, such as alterations in root exudates, soil conditions, and other factors [70]. Therefore, the changes in the rhizosphere community under drought stress still need to be discussed. Additionally, AMF forms a symbiotic relationship with plant roots, aiding plants in coping with environmental cues, especially drought stress [20,71,72]. AMF extend their hyphae into the soil, increasing the surface area for nutrient absorption and accessing water in deeper soil layers [20]. This relationship is crucial for plants to manage drought stress effectively. Future studies should aim to develop a comprehensive understanding of the rhizosphere microbiome by incorporating internal transcribed spacer (ITS) sequencing to analyze fungal communities. 

Drought indirectly influences plant growth by altering soil water availability, nutrient content, soil microbial biomass, and enzyme activity [73,74]. Previous studies have found that the activities of soil sucrase, urease, and phosphatase are influenced by the contents of total nitrogen, organic matter, total phosphorus, and organic matter [75]. Our findings indicated that, under drought-stress conditions, S-SC, S-UE, and S-CAT were positively correlated with soil nutrient levels but not with soil pH (Figure 8). Different crops have different responses to drought, and there can be some differences in soil nutrient deficiency. Previous studies showed that drought stress reduces TN in the soil, slows down the degradation of organophosphorus, and affects the absorption and release rates of carbon by plants [76,77]. Wang et al. [78] pointed out that reduced precipitation can significantly inhibit nutrient supply, microbial growth efficiency, and the decomposition of organic matter. Our study found significant differences in nutrient composition in the rhizosphere soils of cassava varieties with different drought tolerances. Compared to well-watered conditions, drought treatment significantly reduces the contents of total nitrogen, total phosphorus, available phosphorus, and total organic carbon in the rhizosphere soil of both cassava varieties. This observation was mirrored by Liu et al. [79], who reported on a certain genotype-dependent reduction in total nitrogen and available phosphorous in sugarcane under drought stress. Soil microorganisms, the main drivers of soil nutrient cycling, are influenced by the interaction between crops and the soil environment. Moreover, rhizosphere bacteria play crucial roles in stimulating the decomposition of organic matter; enhancing the solubility of C, N, and P; and promoting the growth of host plant roots [80,81]. Drought stress was found to reduce the activities of various soil enzymes, such as S-ACP, S-UE, and enzymes involved in the nitrogen cycle, thereby diminishing the nutrient supply of plants [82]. Enzymes closely related to rhizosphere bacteria, including S-CAT, S-ACP, and S-UE, are key to the nitrogen and phosphorus cycles and soil protein decomposition [83]. This explains the significant correlation between drought-responsive flora and soil physicochemical properties revealed by our redundancy discriminant analysis (RDA). The nutrient deficiency induced by drought is initiated by a decrease in the enzyme cycle rate, responsible for nutrient cycling and decomposition under drought stress [84]. Rhizosphere microorganisms counteract this deficiency by enhancing metabolic activity within the rhizosphere soil environment, thereby decomposing organic matter and promoting the effective mineralization of soil nutrients [85]. Moreover, the changes in the bacterial communities within the cassava rhizosphere were significantly correlated with soil nutrient and soil enzyme activity, which is consistent with Lan et al. [86] and Yuan et al. [87]. The drought-responsive Acidobacteriae of SC124 were positively correlated with S-ACP and S-CAT, whereas Actinobacteria, which were the dominant rhizosphere bacteria of SC8 under drought stress, were positively correlated with S-SC and S-UE. These findings highlight the crucial roles of S-ACP and S-UE in the oxidative metabolism of soil nutrients, impacting the mineralization of phosphorus in the rhizosphere soil and the transformation of soil protein components [88,89,90]. Given the close correlation between rhizosphere environment components, it is evident that changes in the rhizosphere exudates in response to soil nutrient status under water stress significantly affect the response of drought-resistant rhizosphere bacteria [27,91]. Consequently, we concluded that the bacterial community of drought-tolerant cassava varieties exhibits greater stability under drought stress, in contrast to the less stable rhizosphere bacterial community of drought-sensitive varieties, a stability that is significantly correlated with soil nutrient and enzyme activities.

Durán et al. [92] found that the complexity of plant root microbial networks is crucial for plant survival, emphasizing the key role of rhizosphere microflora interactions in the environmental adaptability of host plants. Thus, the observed difference in the rhizosphere bacterial community between the two varieties underscores a significant factor contributing to their variance in drought tolerance. However, host plant resistance to environmental stressors, including drought, also significantly influences the structure of the rhizosphere bacterial community [5]. This symbiotic relationship is underscored by the simultaneous impact of abiotic environmental factors on both plant and microbial community dynamics [93]. Our experimental results showed significant differences between drought-tolerant SC124 and drought-sensitive SC8 under drought stress, not only in the physiological characteristics indicative of drought resistance but also in the responsiveness of core drought-resistant bacteria. Furthermore, environmental factors were found to correlate with both plant physiology and community composition, as demonstrated by the significant positive correlation between the Richness index and parameters such as Fv/Fm, soluble sugars, total nitrogen, and total phosphorus. The Shannon and Simpson indices were positively correlated with total phosphorus and available phosphorous, illustrating how different environmental conditions precipitate distinct shifts in microbial communities, which in turn can further influence the biogeochemical reactions of various substances [94]. An analysis of the relative abundance of 16S rDNA in the rhizosphere revealed similar species compositions but significantly different relative abundances of bacterial strains between the drought-tolerant SC124 and the drought-sensitive SC8 (Figure 5). This observation aligns with a previous study by Kalinowski et al. [95], which posited that plant adaptation to stresses could alter the bacterial composition of the rhizosphere, a hypothesis confirmed by our experiments. In our study, Actinobacteria and Gamma-proteobacteria emerged as the dominant bacterial classes under drought conditions in the drought-sensitive variety SC8, whereas these bacteria were already prevalent in the drought-resistant variety SC124 under well-watered conditions. We deduce that rhizosphere bacteria displaying an increased relative abundance under drought stress are typically those with a certain degree of drought tolerance, capable of maintaining the rhizosphere environment under stress [27,96]. Actinobacteria, a Gram-positive bacterium with a thick cell wall and the capacity to form spores and resist certain stresses, likely dominate in arid environments due to their inherent stress resistance. In addition, Actinobacteria are mostly saprophytic bacteria that thrive during drought and other stressful conditions, are implicated in soil structure formation, and can contribute to crop growth and development [7,97]. Naylor and Coleman-Derr [27] reported that the observed changes in the soil bacteria under drought conditions are typically related only to the alterations in drought-sensitive bacterial species, rather than overall bacteria changes. Vurukonda et al. [98] posited that host plants’ differential drought tolerance capabilities lead to distinct changes in rhizosphere microorganisms under drought stress. Similarly, the observed shifts in Gamma-proteobacteria and Bacteroidia abundance were positively related to drought stress, whereas the increase in the relative abundance of Alpha-proteobacteria was positively correlated with the drought tolerance attributes of the drought-tolerant variety SC124, indicating that these bacteria possess a level of drought resistance. Alpha-proteobacteria, known as plant growth-promoting bacteria, enhance host plant adaptation under drought stress through utilizing the C in the rhizosphere exudates of sugarcane [99]. Therefore, the bacteria recruited by drought-sensitive varieties after stress have been enriched in the drought-resistant varieties, and the types of microorganisms recruited are also different with different drought resistance. Understanding the initial state of the soil microbial community will enhance our ability to develop strategies for managing soil health and optimizing plant–microbe interactions. Bulgarelli et al. [100] demonstrated that the composition of the root microbiome is strongly influenced by the bulk soil microbiota, indicating that initial soil conditions are critical determinants of rhizosphere microbial communities. Similarly, Xu et al. [7] found that drought conditions significantly delayed the development of the *Sorghum bicolor* root microbiome and enriched the specific bacterial taxa, emphasizing the importance of initial soil microbial communities. Thus, future studies should incorporate comprehensive analysis of both bulk soil and rhizosphere microbial communities to fully comprehend how drought stress conditions affect rhizosphere microbial dynamics in cassava. 

Previous studies have demonstrated the importance of microbial diversity for system functioning [101,102,103]. The highest abundance of bacteria in cassava was related to chemoheterotrophy and aerobic chemoheterotrophy, indicative of a close relationship with plant metabolism. The variations in root exudates elucidate how these cassava varieties navigate their rhizosphere bacterial ecosystems, incorporating both core and cultivar-specific microbiota [104]. Drought stress can reduce soil microbial metabolism, but drought-resistant cassava can resist drought by recruiting some nitrogen fixation and nitrate reduction bacteria, which is a consistent mechanism to resist stress. Nitrogen is an important component of nucleotides, proteins, biofilms, and chlorophyll [105,106], and is one of the major plant macronutrients. Drought stress can reduce nitrogen content in soil and Microbialbiomassnitrogen (MBN) cannot be directly absorbed, but microorganisms can convert it into a form that plants can absorb. Numerous PGPRs possess physiological functions like phosphorus dissolution, potassium dissolution, nitrogen fixation, and iron dissolution, directly supplying accessible nutrients to plants. For instance, under drought-stress conditions, the inoculation of *Azospirillum* significantly increased wheat yield per plant and augmented seed nitrogen and phosphorus contents [107]. Therefore, the results indicated that plants secreted inter-root secretions that were involved in a “call for help” strategy and actively engaged their microbes to maximize their survival and growth when affected by external stresses, leading to an enrichment of beneficial bacteria that become essential members of the dominant network [108], and these are also closely related to the characteristics of the plant.

## 4. Materials and Methods

### 4.1. Experimental Design

Two cassava varieties were used in this study: SC8, which is drought-sensitive, and SC124, which is known for its relative drought tolerance [40,109]. A pot culture experiment was conducted at Hainan University, Haikou, Hainan, China (20°50′ N and 108°38′ E), from 2 April to 18 May 2023. Cuttings of about 15 cm from the current year, obtained from ten mother plants per variety, were planted in plastic pots (33 cm upper diameter, 24 cm lower diameter, and 27 cm height; one cutting per pot) filled with a mix of vegetative soil and vermiculite (3:1 ratio) under a rainproof shed under natural conditions. The seedlings were grown under 16/8 h light/dark and 28 °C/22 °C day/night cycles with approximately 60% relative humidity. After planting the cassava cuttings in the pots, we watered them daily to maintain adequate soil moisture content. Additionally, we provided 100 mL of 1/2 Hoagland nutrient solution every three days to supply necessary nutrients to the cassava plants until treatment began. To eliminate any positional effects, we randomly rearranged the pots every 2–3 days. One month after planting, uniform seedlings from each variety were designated into drought (D) and well-watered (C) groups, with the D group experiencing halted watering to achieve a field water capacity of 30 ± 5% (soil volume water content 3.67–4.57%), while the C group maintained a field water capacity of 70 ± 5% (soil volume water content 24.58–24.77%) [110]. Each treatment included at least 12 seedlings. During the drought period, we irrigated the plants using distilled water every two days, following the weighing method [49]. Drought stress treatment lasted for 21 days. Chlorophyll fluorescence decreased under a soil water deficit, and this was more pronounced in plants with certain genotypes [111]. Hence, in this study, leaf chlorophyll fluorescence was used to indicate the level of drought stress. SPAD and Fv/Fm values of three leaves per plant were determined using a SPAD-502 Plus portable chlorophyll meter (Konica Minolta, Tokyo, Japan) and a FluorPen FP100 (FluorPen FP100, Photon Systems Instruments, Prague, Czech Republic), respectively.

### 4.2. Soil Sampling and Physicochemical Assessment

Initial soil and rhizosphere soil samples were collected following a standardized protocol [112] with three biological replicates per treatment, resulting in fifteen soil samples, named Initial soil, SC124-C, SC124-D, SC8-C, and SC8-D. In brief, the root system was excavated along with the surrounding soil. Large soil clumps and stones were removed, and loose soil was gently shaken off. The rhizosphere soil adhering to the root surface was collected with a brush and then sieved through a 2 mm sieve. After sampling, each soil sample was divided into two sub-samples. One sub-sample, weighing about 20 g, was air-dried and stored at 4 °C for subsequent analysis of its physical and chemical properties. The other sub-sample, weighing about 2 g, was frozen in liquid nitrogen and stored at −80 °C for later extraction of rhizosphere microbial DNA. The soil pH was measured using a pH meter S400-K. The content of total organic carbon was determined with potassium dichromate titration. The nitrogen content, including total nitrogen, was determined using the Kjeldahl procedure. Total phosphorus and available phosphorus were determined using UV–Vis spectrophotometry. S-CAT, S-ACP, S-UE, and S-SC activity were detected by using an assay kit (Sangon Biotech Co., Ltd., Shanghai, China) following the manufacturer’s instructions. 

Prior to harvesting, plant height (PH) and stem diameter (SD) were recorded, and soil volume water content was measured using a YT-TS soil moisture meter (Shandong Yuntang Intelligent Technology Co. Ltd., Shandong, China). Biomass was calculated post-harvest, and the root–shoot ratio was determined. The fresh leaves were taken and stored at −80 ℃ for subsequent index determination. The malondialdehyde (MDA), proline content, and soluble sugar content were detected by using an assay kit (Beijing Solarbio Science & Technology Co., Ltd., Beijing, China) following the manufacturer’s instructions.

### 4.3. DNA Extraction and Sequencing

Total DNA was extracted from 1 g of soil using a TGuide S96 Magnetic Soil/Stool DNA Kit (Tiangen Biotech Co., Ltd., Beijing, China) according to the manufacturer’s instructions. The concentration of the extracted DNA was measured using a Qubit dsDNA HS Assay Kit and Qubit 4.0 Fluorometer (Invitrogen, Thermo Fisher Scientific, Waltham, MA, USA). The V3-V4 region of the 16S rDNA gene was amplified from the genomic DNA of each sample using primers F338 (5′-ACTCCTACGGGAGGCAGCA-3′) and R806 (5′-GGACTACHVGGGTWTCTAAT-3′). Both primers were tailed with sample-specific Illumina index sequences for deep sequencing. PCR amplification was conducted in 25 μL reaction volumes containing 5 μL of Q5 Reaction Buffer, 200 μM of dNTP Mix, 0.5 μM of each primer, 0.02 U μL^−1^ Q5 High-Fidelity DNA Polymerase (New England Biolabs, Hitchin, UK), and approximately 10 ng of DNA template. The thermal cycling conditions were as follows: initial denaturation at 94 °C for 5 min, followed by 30 cycles of denaturation at 95 °C for 30 s, annealing at 50 °C for 30 s, and extension at 72 °C for 40 s, with a final elongation at 72 °C for 7 min. PCR amplicons were purified using Agencourt AMPure XP Beads (Beckman Coulter, Indianapolis, IN, USA) and quantified with a Qubit dsDNA HS Assay Kit (Thermo Fisher Scientific, MA, USA) and Qubit 4.0 Fluorometer (Invitrogen, Thermo Fisher Scientific, MA, USA). The purified amplicons were pooled in equal amounts for sequencing. Sequencing was performed by Biomarker Technologies Co., Ltd. (Beijing, China) using the Illumina HiSeq 6000 platform (Illumina, Santiago, CA, USA). The raw Illumina sequences of the rhizosphere soil samples were deposited in the Sequence Read Archive (SRA) under project PRJNA1098786 “https://www.ncbi.nlm.nih.gov/sra/PRJNA1098786” (accessed on 13 April 2024). Pairs of reads from the original DNA fragments were merged using FLASH software version 1.2.11 [113].

### 4.4. Processing of Sequence Data 

Paired amplicon sequencing reads were joined using FLASH version 1.2.11 [113], quality-filtered using Trimmomatic version 0.33 [114], and demultiplexed. Chimeric reads were identified and removed using UCHIME version 8.1 [115]. The remaining high-quality reads were clustered into operational taxonomic units (OTUs) at a ≥97% sequence similarity threshold using USEARCH version 10.0 [116]. OTUs with abundances less than 0.005% of the total sequences were excluded. Taxonomic annotation of the OTUs was performed using the Naive Bayes classifier in QIIME2 [117] with the SILVA database [118] and a confidence threshold of 70%.

### 4.5. Statistical and Bioinformatics Analysis

The species diversity of each sample was analyzed based on four alpha (*α*) diversity indices: ACE, Chao1, Shannon, and Simpson. All sample indices were calculated and displayed by QIIME2 “https://qiime2.org/” (accessed on 5 April 2024), respectively. A Kruskal–Wallis test followed by a least significant difference (LSD) test were performed to test the significance of the effects of drought and variety on soil physicochemical property using SPSS 23.0 (IBM Corp., Armonk, NY, USA). Beta (*β*) diversity analysis was used to evaluate differences in terms of species complexity, and principal coordinate analysis (PCoA) based on Weighted and Unweighted UniFrac distance was calculated and plotted using the *R* package version 3.6.3 to determine the main variable components in the rhizosphere bacterial communities [119]. Venn diagrams showing shared and specific OTUs between different groups were also constructed. Linear discriminant analysis effect size (Lefse), for which the logarithmic LDA score was set to 4.0 with statistical significance (*p* < 0.05), and the FAPROTAX functional prediction of the bacterial community were performed on the BMK Cloud platform “www.biocloud.net” (accessed on 2 April 2024). The relationship between bacterial communities and environmental factors and plant morphological and physiological were analyzed using the Mantel test [120]. We computed pairwise distances between each environmental factor using the ggcor package version 3.6.2 in *R*. To identify the relationship between microbial community composition and environmental factors, Mantel correlations were computed (9999 permutations) using the ggcor package in *R* [120]. Redundancy discriminant analysis (RDA) was carried out using software Canoco version 5.0.

## 5. Conclusions

In summary, our research drew a summary map of the response of cassava rhizosphere microorganisms to drought stress, showcasing significant differences in plant height, physiological parameters, soil nutrients, and soil microbiological traits between two cassava varieties with contrasting drought resistance under drought treatment compared to well-watered conditions (Figure 10). Drought-tolerant varieties of SC124 actively modulate their internal environment to enhance adaptability to drought by increasing the MDA and proline content, while reducing the soluble sugar content to regulate osmotic pressure and maintain soil nutrient balance and soil enzyme activity. Conversely, the drought-sensitive varieties of SC8 experience a reduction in core microorganisms under drought stress, with a severe restriction in bacterial species interactions. The rhizosphere bacterial community of the drought-tolerant variety SC124, enriched with core microorganisms, ensures the stability of the microbial niche under adverse conditions. Under drought treatment, the cassava was dominantly enriched with beneficial microorganisms, such as Actinobacteria, Alpha-proteobacteria, Gamma-proteobacteria, Acidobacteriae, Polyangia, and other bacteria, which play crucial roles in enhancing the drought resistance of cassava. Overall, our findings support the enrichment of differential beneficial microorganisms in both drought-tolerant cassava varieties (SC124) and drought-sensitive varieties (SC8), suggesting that these beneficial microorganisms possess the capability to assist crops in mitigating drought damage.

## Figures and Tables

**Figure 1 ijms-25-07326-f001:**
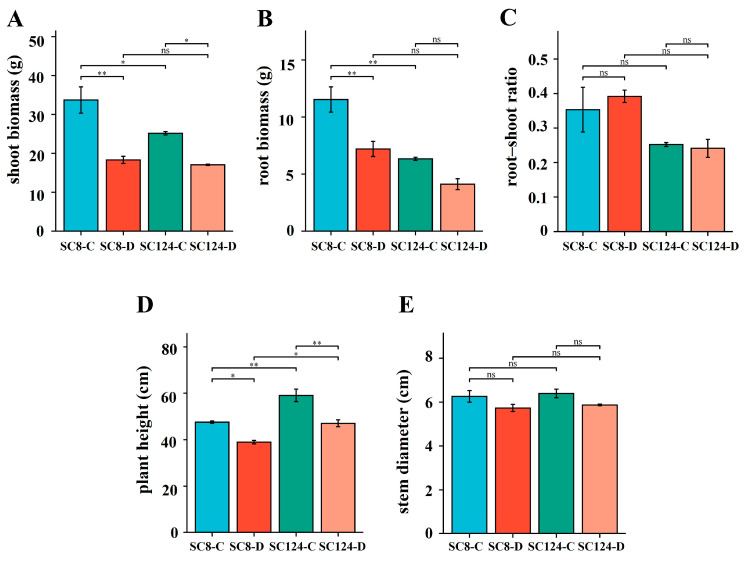
Morphology of different varieties of cassava under drought stress. (**A**–**E**) Shoot biomass, root biomass, root–shoot ratio, plant height, stem diameter. SC124-C: drought-tolerant cassava under well-watered conditions, SC124-D: drought-tolerant cassava under drought conditions, SC8-C: drought-sensitive cassava under well-watered conditions, SC8-D: drought-sensitive cassava under drought conditions. Student’s *t*-test was used to analyze the significant difference between the two varieties under drought stress. Data are the means of three replicates, and error bars indicate standard deviations. Asterisks indicate statistical significance, * *p* < 0.05, and ** *p* < 0.01; “ns” means non-significant difference.

**Figure 2 ijms-25-07326-f002:**
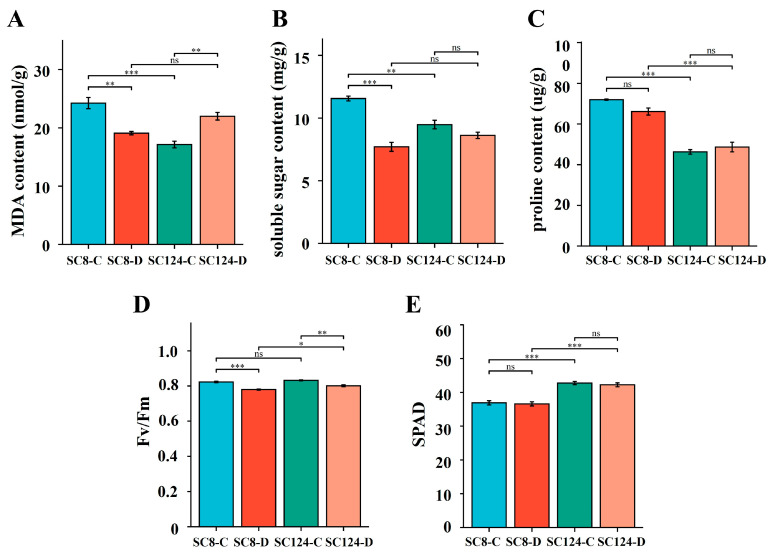
Physiological and photosynthetic characteristics of different varieties of cassava under drought stress. (**A**–**E**) MDA content, soluble sugar content, proline content, Fv/Fm, SPAD. SPAD, relative chlorophyll content, Fv/Fm, quantum photochemical efficiency of photosystemⅡ. SC124-C: drought-tolerant cassava under well-watered conditions, SC124-D: drought-tolerant cassava under drought conditions, SC8-C: drought-sensitive cassava under well-watered conditions, SC8-D: drought-sensitive cassava under drought conditions. Student’s *t*-test was used to analyze the significant difference between the two varieties under drought stress. Data are the means of three replicates and error bars indicate standard deviations. Asterisks indicate statistical significance, * *p* < 0.05, ** *p* < 0.01, and *** *p* < 0.001; “ns” means non-significant difference.

**Figure 3 ijms-25-07326-f003:**
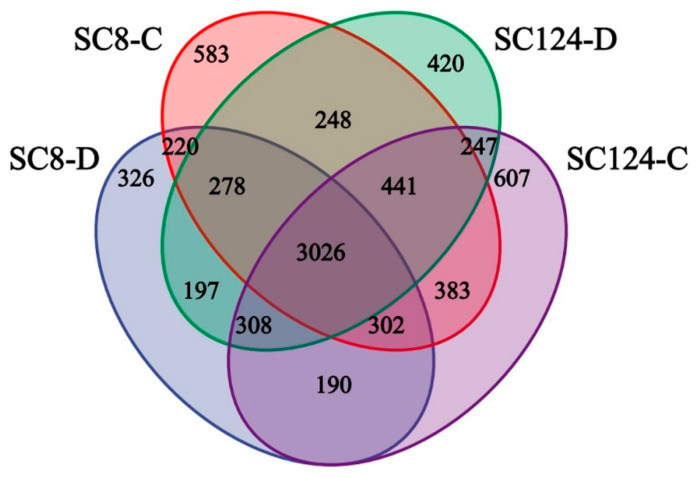
Venn diagram of bacterial OTU distribution in rhizosphere soil of different cassava varieties under drought stress. SC124-C: drought-tolerant cassava under well-watered conditions, SC124-D: drought-tolerant cassava under drought conditions, SC8-C: drought-sensitive cassava under well-watered conditions, SC8-D: drought-sensitive cassava under drought conditions. The overlapped part indicates a common OTU, and the un-overlapped part indicates a unique OUT.

**Figure 4 ijms-25-07326-f004:**
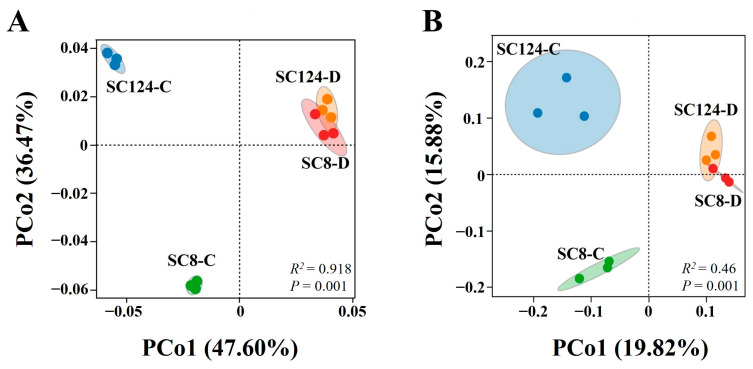
PCoA analysis of *β* diversity based on Unifrac distance. (**A**) PCoA analysis of *β* diversity based on unweighted UniFrac distance, where PCo1 axis shows the effect of water treatment, and PCo2 represents the effect of variety differences on microbial community *β* diversity. (**B**) PCoA based on weighted UniFrac distance according to the analysis; the PCo1 axis shows the effect of water treatment and the PCo2 represents the effect of variety differences on the microbial community *β* diversity. SC124-C: drought-tolerant cassava under well-watered conditions, SC124-D: drought-tolerant cassava under drought conditions, SC8-C: drought-sensitive cassava under well-watered conditions, SC8-D: drought-sensitive cassava under drought conditions.

**Figure 5 ijms-25-07326-f005:**
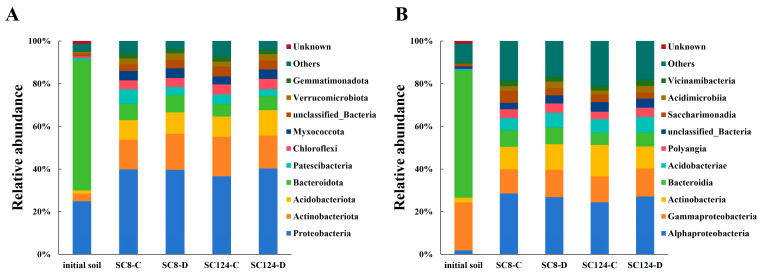
Differences in the composition of bacterial communities in different water conditions and varietal treatments. (**A**) The relative abundance of the bacterial community on the phylum level in different water and varietal treatments. (**B**) The relative abundance of the bacterial community on the class level in different water and varietal treatments. SC124-C: drought-tolerant cassava under well-watered conditions, SC124-D: drought-tolerant cassava under drought conditions, SC8-C: drought-sensitive cassava under well-watered conditions, SC8-D: drought-sensitive cassava under drought conditions.

**Figure 6 ijms-25-07326-f006:**
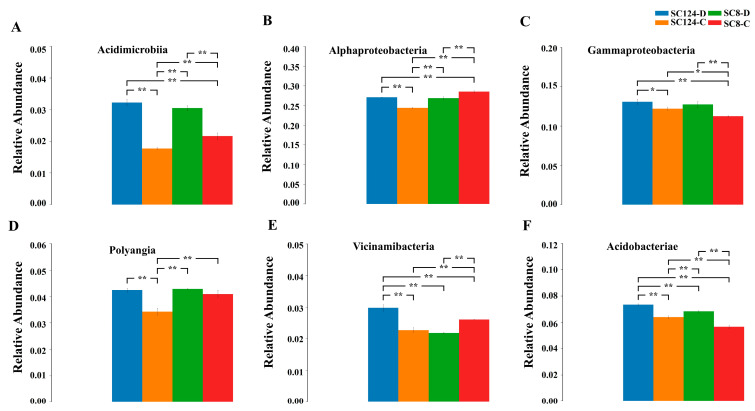
Variance analysis of different varieties of cassava soil microorganisms under drought stress based on bacterial abundance difference at class level. (**A**) Acidimicrobiia. (**B**) Alphaproteobacteria. (**C**) Gammaproteobacteria. (**D**) Polyangia. (**E**) Vicinamibacteria. (**F**) Acidobacteriae. Data are the means of three replicates, and error bars indicate standard deviations. Asterisks indicate statistical significance, * *p* < 0.05, and ** *p* < 0.01.

**Figure 7 ijms-25-07326-f007:**
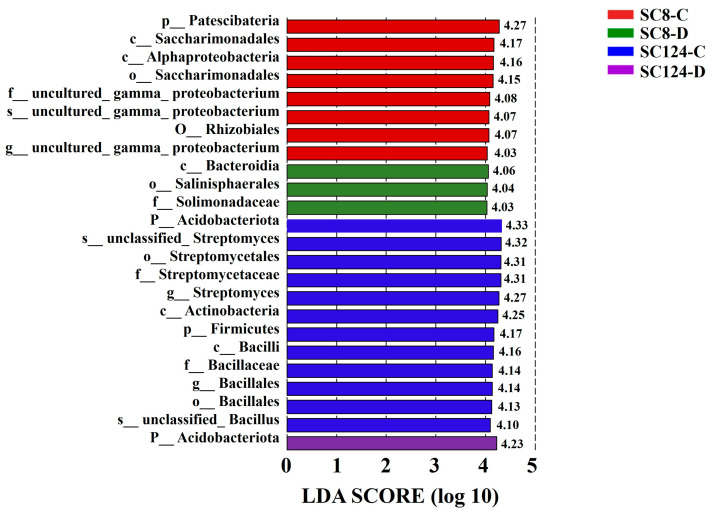
Linear discriminant analysis of effect size (LEfSe) showing differentially abundant genera within drought-treated versus well-watered rhizosphere soil of the two cassavas based on a cut-off of *p* < 0.05 and an LDA score > 4.0. LDA: linear discriminate analysis. SC124-C: drought-tolerant cassava under well-watered conditions, SC124-D: drought-tolerant cassava under drought conditions, SC8-C: drought-sensitive cassava under well-watered conditions, SC8-D: drought-sensitive cassava under drought conditions.

**Figure 8 ijms-25-07326-f008:**
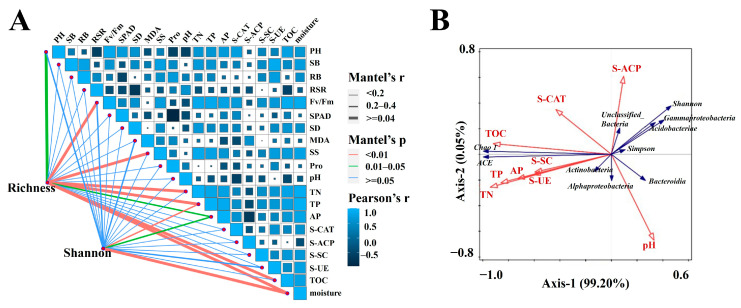
Correlation analysis of environmental factors with plant morphology and physiology, community abundance, and diversity. (**A**) Correlations between environmental factors and richness and Shannon indices in the rhizosphere of cassava. Line width is proportional to the partial Mantel’s r value, and line color denotes the statistical significance based on 999 permutations (orange, *p* < 0.05; green, 0.01 < *p* < 0.05; blue, *p* ≥ 0.05). Pairwise comparisons of environmental factors are also shown, with a gradient of dark blue and light blue colors denoting Pearson’s correlation coefficients. *p* values were adjusted for multiple testing with the Holm–Bonferroni method. (**B**) Redundancy discriminant analysis of soil physicochemical properties and cassava rhizosphere bacterial community under drought stress based on OUT. TOC, total organic carbon. AP, available phosphorous. TN, total nitrogen. TP, total phosphorus. S-CAT, soil-catalase. S-ACP, soil-acid phosphatase. S-UE, soil urease. S-SC, soil saccharase. PH, plant height. SB, shoot biomass. RB, root biomass. RSR, root–shoot ratio. SD, stem diameter. MDA, malondialdehyde. SS, soluble sugar. Pro, proline.

**Figure 9 ijms-25-07326-f009:**
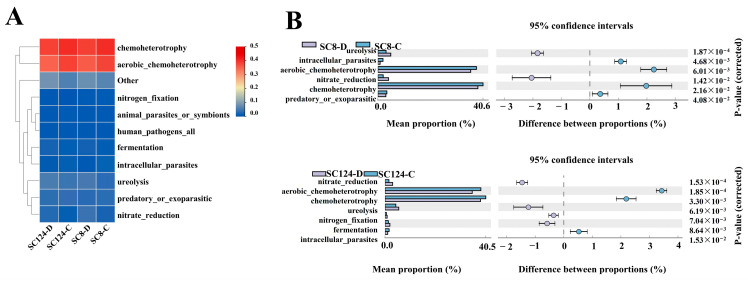
Prediction of FAPROTAX function of rhizosphere soil bacteria in different cassava varieties under drought stress. (**A**) Heat map of FAPROTAX function prediction. (**B**) Histogram of differences in FAPROTAX function prediction; *p* < 0.05. SC124-C: drought-tolerant cassava under well-watered conditions, SC124-D: drought-tolerant cassava under drought conditions, SC8-C: drought-sensitive cassava under well-watered conditions, SC8-D: drought-sensitive cassava under drought conditions.

**Figure 10 ijms-25-07326-f010:**
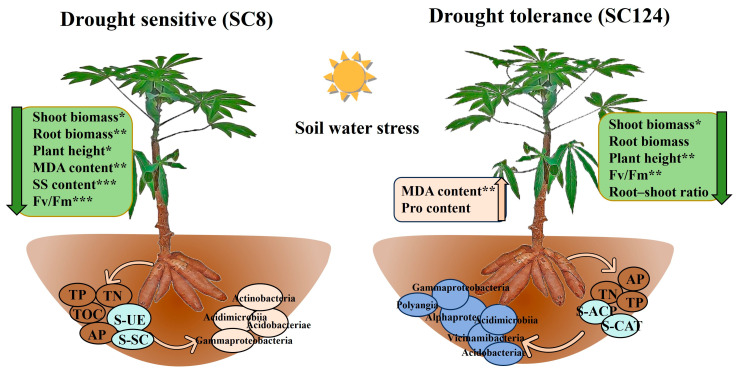
Summary diagram of the response mechanism of cassava rhizosphere to drought stress. TOC, total organic carbon. AP, available phosphorous. TN, total nitrogen. TP, total phosphorus. S-CAT, soil-catalase. S-UE, soil urease. S-SC, soil saccharase. S-ACP, soil acid phosphatase. MDA, malondialdehyde. SS, soluble sugar. Pro, proline. * *p* < 0.05, ** *p* < 0.01, and *** *p* < 0.001.

**Table 1 ijms-25-07326-t001:** Physical and chemical properties of soil under different treatments.

		Soil Volume Water Content (%)	pH	TN (mg/g)	TP (mg/g)	AP (mg/g)	TOC (%)	S-CAT (U/g)	S-ACP (μmol/g)	S-SC (U/g)	S-UE (U/g)
SC8	Control	24.77 ± 1.66 a	5.71 ± 0.07 a	1.54 ± 0.13 a	0.85 ± 0.01 a	18.74 ± 1.04 a	1.52 ± 0.02 a	5.94 ± 0.39 ab	48.54 ± 4.81 a	6.21 ± 0.83 a	2.70 ± 0.35 a
	Drought	3.67 ± 0.31 b	5.85 ± 0.34 a	0.72 ± 0.08 c	0.79 ± 0.01 b	16.25 ± 0.42 b	1.09 ± 0.12 b	5.36 ± 0.53 b	47.49 ± 3.30 a	4.43 ± 1.12 a	1.98 ± 0.17 bc
SC124	Control	24.58 ± 0.83 a	5.50 ± 0.26 a	1.41 ± 0.07 a	0.85 ± 0.01 a	20.40 ± 0.92 a	1.52 ± 0.02 a	7.26 ± 1.26 a	44.20 ± 3.38 a	6.18 ± 0.66 a	2.36 ± 0.25 ab
	Drought	4.57 ± 0.05 b	5.58 ± 0.14 a	0.98 ± 0.08 b	0.79 ± 0.01 b	16.87 ± 0.65 b	1.35 ± 0.10 a	5.37 ± 0.51 b	52.39 ± 1.57 a	4.64 ± 1.16 a	1.70 ± 0.11 c
Kruskal-Wallice Test											
	H	9.495	2.077	9.667	8.687	8.333	9.012	3.923	6.897	4.385	8.744
	*p*	0.023	0.557	0.022	0.034	0.040	0.029	0.270	0.075	0.223	0.033

Values are mean of three soil samples ± SD. Different letters indicate significant differences (ANOVA, *p* < 0.05, Tukey’s HSD post hoc analysis, *p* < 0.05) among treatments. TN, total nitrogen; TP, total phosphorus; AP, available phosphorous; S-CAT, solid catalase; S-ACP, solid acid phosphatase; S-SC, soil saccharase; S-UE, soil urease; TOC, total organic carbon. Significant at 0.001 level. SD, standard deviation. H: test statistics; *p*: progressive significance.

**Table 2 ijms-25-07326-t002:** Analysis of *α*-diversity in different varieties of cassava rhizosphere soil under drought stress.

Treatment	ACE Index	Chao1 Index	Simpson Index	Shannon Index
SC124-C	4464 ± 15 b	4403 ± 28 b	0.991 d	9.175 ± 0.019 d
SC124-D	4184 ± 25 c	4150 ± 31 c	0.995 a	9.497 ± 0.014 a
SC8-C	4606 ± 22 a	4524 ± 23 a	0.992 c	9.287 ± 0.007 c
SC8-D	3786 ± 20 d	3757 ± 20 d	0.995 b	9.338 ± 0.030 b

Results are reported as means ± SD (n = 3). Different lowercase letters between the same column indicate significant differences (*p* < 0.05). SD, standard deviation.

## Data Availability

The original contributions presented in the study are included in the article/Appendix A, further inquiries can be directed to the corresponding authors.

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
