# Peer review of "Influence of Drought Stress on the Rhizosphere Bacterial Community Structure of Cassava (Manihot esculenta Crantz)"

_ijms, 2024, doi:10.3390/ijms25137326_

Round 1

Reviewer 1 Report (New Reviewer)

Comments and Suggestions for Authors

The manuscript contains original and scientific information regarding the impact of drought and genotype (two cassava (Manihot esculenta Crantz) varieties have been used: SC8 - drought-sensitive, and SC124 drought tolerant) on the diversity and composition of rhizosphere bacterial communities; identifying bacterial taxa linked to increased drought stress tolerance; and explore the relationship between bacterial community structure and soil environmental factors.

The manuscript is clear, relevant to the scientific field of the Journal and presented in a well-structured manner.

The cited references are mostly recent publications (within the last 5 years - 46%) and are appropriately selected. The list of References does not include an excessive number of self-citations.

In the section Material and Methods - For what period (how long) the plants are subjected to drought?

Separate this subsection “Soil Sampling and Physicochemical Assessment” into two parts. First - describe soil sampling and biochemical analysis; Second - describe plant sampling and biochemical analysis

The supplementary table 1 is superfluous. It is not necessary to present the raw data on soil's physical and chemical properties.

In the section “Discussion” could you describe more in detail why the well-watered SC124 variety has lower shoot biomass but higher plant height than the well-watered SC8 variety? The results are presented in Figure 1.

All remarks are made in the text.

The manuscript could be accepted for publication after minor revision.

Comments on the Quality of English Language

The English language is appropriate and understandable.

Author Response

Reviewer 2 Report (New Reviewer)

Comments and Suggestions for Authors

Authors provide interesting study, in the field that is often not taken into account. But I am affraid, that results are quite incomplete - I understand, they wanted to focus on bacteria, but fungi (AMF) are of same importance, especially when talking about drought stress.

Authors should at least discuss why they ommited fungi from their experiments.

Authors state, that plants were wattered by sterile distilled water. This itself can be sort of stress, and also affect the soil conditions in context of nutrients, elementary composition etc. and therefore even the microbial composition. I would suggest some mild buffer as dilluted Knopps solution, or something with at least some ionic strenght, especially if autors conclude osmotic pressure is one of the main answers (line 630). Authors should present data, that would show, that this is not the case - at least the nutrients levels with same soil watered in their used scheme.

Authors often use the word "genotype" in confusing manner. I understand what they are trying to say, but it is rather imprecise. I would suggest rephrasing most of the senteces. Use variety where possible, or "plants of certain genotype" - genotype itself is not doing anything.

Above mentioned are my main concerns, below are some other minor comments (some of them repeated from above).

line 42 not only bacterial

lines 66/67 the genotype is a misleading word here, same in 79/80

Names of enzymes should be stated in full when first used (lines 111, 113). Some context of the enzyme activity levels should be given. Also these lines are hard to understand, was the activity of both enzymes higher in the SC124? It doesnt correspond with table 1.

Does the pH and N/P content in soil affect activities of measured enzymes? If so, was this somehow taken into account?

Why are not all of the values from table one discussed in the result section (e.g.? S-SC, S-ACP)

Results from lines 164-171 are not as interesting as these characteristics of the sequencing but not the merit of the results I would move to SI. Furthermore I cannon fully agree with the conclusion taken from them - if there would be some higly abundant bacteria, you cannot conclude your statemes just from these numbers.

I would really not use some many digits in table 2 (if your SD is 15, why go to thousands of units?)

In figure 7 I would suggest to focus on differences meaning cutting the LDA score from 0-3 and extended the 3+ region.

Comments on the Quality of English Language

Some mentioned above. Generally checking of tenses should be done.

Round 2

Reviewer 2 Report (New Reviewer)

Comments and Suggestions for Authors

I hope authors found my advices useful, and we can both agree that the manuscript is better now, and ready for publication.

This manuscript is a resubmission of an earlier submission. The following is a list of the peer review reports and author responses from that submission.

Round 1

Reviewer 1 Report

Comments and Suggestions for Authors

This manuscript present instigation of the drought stress on cassava plant and it microbiome. The introductions present sufficient background information for the presented results, and discussion refers to the relevant publication, showing justified conclusions. The methods are properly selected and described. I wish the authors would also include at least one additional housekeeping gene in the analysis, but otherwise i see no major issues. The results are clearly presented. I do appreciate the number of analytical approaches including taking into account 4 diversity indexes. This shows that the authors are aware of the indexes weakness in the microbial community analysis and suggest deep understanding of the topic. In my opinion this is a high quality research that deserves to be published in International Journal of Molecular Sciences. In conclusion I recommend this manuscript to be accepted for publication after minor revisions. Please look bellow for more detailed comments.

Line 112: Were the masses adjusted to water content of soil?

Line 114: Please be aware that this type of statistic is not suitable for such kind of data. I understand that you would like to check both factors and the factors combined, and therefore this is probably the easiest method. Therefore I think such analysis is acceptable in that case, but please add the row data to the supplementary files. Generally ANOVA is a parametric test and the data should meet the assumptions for such test which for 3 replicates is hardly possible. I am aware that the statistical methods are not designed for such kind of data but please consider using more robust test such as Kruskal-Wallice test. For comparing between two sample you could use test for comparing two samples which usually are more sensitive.

Line 132: Please increase the fonts on this figure and the next one

Line 152: Generally extracting information on bacterial taxa abundances is burdened with significant bias. Not only some sequences are better amplified than others, but also different bacteria strains have different copy numbers of 16s rDNA. To tackle that problem I would advise using at least one more housekeeping gene. However if 16s rDNA genotyping is considered insufficient for assessing absolute abundances of microorganisms it still can offer information on the relative abundances and changes please bear that in mind.

Line 227: Please upload the figures with larger font

Line 235: Likewise

Line 346 repetition “suggesting that”

Line 509: Please indicate at lest the sampling time the amount of soil collected

Line 571: p not P

Line 600: This in a very nice summarizing figure, please consider preparing similar figure for the graphical abstract.

Line 601: Please explain used abbreviation in the figure caption.

Reviewer 2 Report

Comments and Suggestions for Authors

In this study, growth and physiological characteristics of drought-tolerant (SC124) and drought-sensitive (SC8) cassava varieties were analyzed under conditions of both well-watered and drought stress. The overall taxonomic structure of the rhizosphere bacterial communities of the two varieties was highly similar.

 The topic in this study was interesting, but definition of rhizosphere bacterial communities was critically unclear. According to the text on L509, “rhizosphere soil samples” were collected for DNA analysis as the same manner of reference [100].  But ref [100] reported how to collect DNA samples from three compartment, “soil”, “rhizosphere”, and “endophytic”. In this study, DNA in which soil or rhizosphere were analyzed? The bacterial community can change with time, why samples on only day 21? Those bacteria were from mother plants and/or vegetative soil/vermiculite? The bacterial community data of both soil (with and without cassava) and rhizosphere on day 0 should be needed. Considering soil heterogeneity, it was difficult to discuss the difference of the taxonomic structure of the bacterial communities.

How was the culture condition of cassava variations, temperature and soil moisture measurement and watering frequency?  For adjusting the water content in soil, water was autoclaved or filtered for removing bacteria ?  

L598-599, it was concluded that these beneficial microorganisms possess the capability to assist crops in mitigating drought damage. Drought condition can change soil bacteria. But evidence for bacterial support on crops this was poor. In abstract, it was written that SC124 mainly relied on Gamma-proteobacteria and Acidobacteriae in its response to drought stress, and the abundance of this class was positively correlated with S-ACP. But Fig. 10, those were depicted as same as Alpha-proteobacteria, Actinobacteria, Polyangia and Vicinamibacteria, Acidimicrobiia. S-UE and S-CAT were highlighted rather than S-ACP. SC8 mainly relied on Actinobacteria in its response to drought stress, and the abundance of this class was positively correlated with soil urease S-UE and S-SC. But Fig. 10, Actinobacteria did not appear and only S-UE.

L104. it was written that S-UE activity in the SC8 rhizosphere soil was higher than that in the SC124 soil under both control and drought conditions. But on Table 1, those were marked by the same letter a, no significant difference.         

L105-106. It was written that the S-CAT activity was higher in the SC124 rhizosphere soil than in the SC8 soil. But on Table 1, actually ab and a for control and b and b for drought, no significant difference.                 

L118-119. It was written that After exposure to drought stress, reductions were observed in shoot biomass, root biomass, plant height, stem diameter, Fv/Fm and SPAD in both varieties.

But Figure 1B, root biomass of SC124-C and SC124D were marked as ns. This should be **. Figure 1E and 2E, stem diameter and SPAD were marked as ns. The text should be wrong.

L123-124. It was written that SC124 exhibited higher plant height, stem diameter, Fv/Fm, and SPAD under both compared to SC8. But Fig. 1E stem diameter were marked as ns.

 L321. Under both the under

L342. Under the same leaf potential Under the same leaf potential

L346 suggesting that suggesting

There are 108 references, but there are generally many vague citations. It would be better to show similar research on rhizosphere bacteria of grains other than cassava in the introduction.

Comments on the Quality of English Language

L321. Under both the under

L342. Under the same leaf potential Under the same leaf potential

L346 suggesting that suggesting

Round 2

Reviewer 2 Report

Comments and Suggestions for Authors

In the revised manuscript, the rhizosphere was clearly defined.  However, the overall taxonomic structure of the rhizosphere bacterial communities of the two cultivars was similar.  Ultimately, bacterial community data on initial conditions and bulk soil were lacking.
